# The Impact of Exercise on Redox Equilibrium in Cardiovascular Diseases

**DOI:** 10.3390/jcm11164833

**Published:** 2022-08-18

**Authors:** Paweł Sutkowy, Joanna Wróblewska, Marcin Wróblewski, Jarosław Nuszkiewicz, Martyna Modrzejewska, Alina Woźniak

**Affiliations:** Department of Medical Biology and Biochemistry, Ludwik Rydygier Collegium Medicum in Bydgoszcz, Nicolaus Copernicus University in Toruń, 85-092 Bydgoszcz, Poland

**Keywords:** redox balance, cardiovascular diseases, physical activity, cardiovascular risk factors, biomarkers

## Abstract

Cardiovascular diseases constitute the most important public health problem in the world. They are characterized by inflammation and oxidative stress in the heart and blood. Physical activity is recognized as one of the best ways to prevent these diseases, and it has already been applied in treatment. Physical exercise, both aerobic and anaerobic and single and multiple, is linked to the oxidant–antioxidant imbalance; however, this leads to positive adaptive changes in, among others, the increase in antioxidant capacity. The goal of the paper was to discuss the issue of redox equilibrium in the human organism in the course of cardiovascular diseases to systemize updated knowledge in the context of exercise impacts on the organism. Antioxidant supplementation is also an important issue since antioxidant supplements still have great potential regarding their use as drugs in these diseases.

## 1. Introduction

Cardiovascular diseases (CVDs) are a group of disorders of the heart and blood vessels. They include coronary heart disease (CHD), cerebrovascular disease, peripheral arterial disease (PAD), rheumatic heart disease (RHD), congenital heart defects (CHDs), deep vein thrombosis (DVT) and pulmonary embolism (PE). Although CVDs are well recognized and, for this reason, effectively treated, they cause the highest number of deaths annually. According to the World Health Organization (WHO), 32% of all global deaths in 2019 were caused by CVDs, mainly heart attack and stroke. Most CVDs can be prevented by addressing behavioral risk factors such as tobacco use, unhealthy diet, physical inactivity and harmful use of alcohol [1]. Physical inactivity, which is a sedentary lifestyle, is a particularly important health problem of modern civilization, as it is the main cause of excess weight and obesity, which are major contributors of morbidity and mortality [2].

One of the most crucial pathogenic factors in the case of CVDs is oxidative stress, which is the disturbed oxidant–antioxidant balance towards an excess of oxidants—reactive oxygen species (ROS) and reactive nitrogen species (RNS). Crucial here are primary endogenous ROS and RNS, superoxide anion radical (^•^O_2_^−^) and nitric oxide (^•^NO) since they are the source of subsequent ROS and RNS. The first one is the product of the reduction of the oxygen molecule by one electron (instead of four) in the mitochondrial transport chain and upregulated activities of oxidoreductases (xanthine oxidoreductase, XOR and nicotinamide adenine dinucleotide phosphate (NADPH) oxidase, NOX). The second one, in turn, is most often produced by endothelial nitric oxide synthase (eNOS). Oxidative stress causes damage (oxidative modification, nitration) to proteins, amino acids, lipids and nucleic acids, which may contribute to inflammation [3] (Figure 1).

A great tool for the prevention and treatment of CVDs may be physical activity (PA) on account of the fact that it improves the antioxidant response against ROS and RNS. Leaving aside many aspects of exercise-induced impacts on the organism, the impact of PA on redox balance seems especially important in the context of CVDs [4]. During physical exercise, oxygen uptake may be even 20 times greater versus the uptake at rest, whereas the oxygen consumption in muscles may increase even 200 times. Thus, the production of ROS increases significantly (the abovementioned incomplete reduction of O_2_ in mitochondria). This is also strictly associated with the intensified generation of RNS and is strongly and positively correlated with exercise intensity [5]. The postexercise oxidative stress may induce radical-mediated microinjuries of muscle fibers and connective tissues, which results in muscle pain, the prolongation of recovery and, consequently, a decrease in sports performance. Thus, it seems reasonable to reduce the exercise-induced ROS and RNS formation in order to improve performance, for example, via antioxidant supplementation [6]. However, there are also reports that indicate a completely opposite effect of such a procedure since it is the fact that reactive oxygen and nitrogen species are necessary for proper muscle contraction both at rest and during PA. Therefore, the literature recommends an appropriate supply of antioxidants in the form of a balanced diet as the best method of maintaining the oxidant–antioxidant equilibrium in exercising people [7]. Besides, a key issue is the adaptation for the increased concentration of reactive oxygen and nitrogen species after exercise because the organism can adapt to them by enhancing its antioxidant response and thus can adapt to exercise-induced disorders. In professional sportsmen, increased activities of antioxidant enzymes are commonly observed [5]. Similarly, in diseases of the cardiovascular system, oxidative stress is a key issue, and antioxidant supplementation has found no clear benefit in the prevention and treatment of CVDs [8,9]. Therefore, the current, popular direction for scientific research is searching for non-pharmacological agents. This primarily applies to physical training, which is included in the WHO recommendations regarding the prevention of CVDs, and positive impacts, which may result primarily from the adaptation of the organism to increased concentrations of reactive oxygen and nitrogen species [10].

The present paper provides an overview of the role of radical-mediated stress in CVDs, the role of PA in their prevention and treatment and, finally, it discusses the effectiveness of antioxidant supplementation in this context.

## 2. The Oxidant–Antioxidant Balance in Human Organisms

In the course of numerous disease entities, ROS and RNS are an etiological factor, and the generation of these molecules increases as a result of homeostasis disorders [11,12]. Disease states are not the only source of ROS and RNS. The aerobic metabolic processes necessary for the proper functioning of tissues pose a great challenge to the mechanisms responsible for maintaining redox homeostasis [13,14]. Reactive oxygen and nitrogen species constitute a highly reactive group of radicals and non-radical chemical compounds which contain oxygen or nitrogen in their structure [15]. These molecules are characterized by a short lifetime due to interactions with other biomolecules present in the cell [16]. ^•^O_2_^−^, hydrogen peroxide (H_2_O_2_) and hydroxyl radicals (^•^OH) are ROS generated by endogenous metabolic processes involving NOX, myeloperoxidase (MPO) and lipoxygenase (LOX) [17]. The main endogenous source of RNS is the metabolism of L-arginine with the participation of selected isoforms of nitric oxide synthase (NOS)—eNOS, inducible NOS (iNOS) and neuronal NOS (nNOS). Reactions catalyzed by these enzymes produce ^•^NO [18]. External environmental factors are also sources of reactive oxygen and nitrogen species. Air pollution related to industrial human activity leads to an increase in the level of sulfur dioxide (SO_2_), carbon monoxide (CO), ozone (O_3_) and nitrogen dioxide (NO_2_) [19]. Additionally, exposure to ionizing radiation, smoking and chronic inflammation lead to a significant increase in ROS generation [20] (Figure 1).

Both ROS and RNS fulfill important biological functions and are essential for the maintenance of homeostasis. ROS are involved in the redox communication between cells [21]. This phenomenon enables the control of cell proliferation and apoptosis [22]. The immune system uses ROS to neutralize pathogens in a process called the respiratory burst [23]. ^•^NO is essential for the proper functioning of the cardiovascular system. This RNS maintains adequate blood flow by regulating the resistance of blood vessels and also inhibits the aggregation of platelets and leukocytes [24].

Reactive oxygen and nitrogen species generated in excessive amounts lead to unfavorable phenomena. Antioxidants are molecules that reduce the negative impact of ROS and RNS on tissues [25]. One of the lines of antioxidant defense are endogenous biomolecules. Antioxidant enzymes such as superoxide dismutase (SOD), catalase (CAT) and glutathione peroxidase (GPx) have a crucial role in this system [26]. Endogenous antioxidants with a low molecular mass are equally important. This group includes glutathione, uric acid, lipoic acid, bilirubin and melatonin [27]. Compounds with antioxidant potential may be supplied with food. Exogenous antioxidants such as vitamins A, C, E and polyphenols are especially important when endogenous systems cease to function effectively [28].

The condition where the generation of ROS and RNS is particularly intense and the antioxidant mechanisms are insufficient is called oxidative stress. Free radicals are reactive with proteins, lipids and the genetic material of the cell [29]. Oxidative modifications of short peptides and large proteins lead to the impairment of the functions of individual biomolecules, affecting the conformation of the protein and the activity of enzymes [30]. After the interaction with ROS, lipids building cell membranes undergo peroxidation and no longer constitute an effective barrier protecting the internal elements of the cell [31]. Mutations and, consequently, carcinogenesis are the result of the influence of ROS on DNA [32]. Considering the negative influence of oxidative stress on biomolecules and, at the same time, the significant role of ROS and RNS, it is necessary to maintain a redox balance.

## 3. Oxidative Stress in Cardiovascular Diseases

Oxidative damage to the cardiovascular system concerns first of all the lipid membranes of the cardiac and vascular myocytes [33] and generates numerous peroxide and aldehyde compounds such as lipid hydroperoxides, malondialdehyde (MDA), F2-isoprostanes and 4-hydroxy-2-nonenal (HNE) [34,35,36,37,38,39,40]. All these compounds are known as the classical biomarkers of oxidative stress (Figure 1).

Since 1968, the level of MDA in biological samples has been measured by means of the spectrophotometric method with thiobarbituric acid (TBA) [41]. Unfortunately, under conditions of this commonly used assay, non-lipid compounds, e.g., derivatives of glucose, sucrose, 2-deoxyribose and sialic acid, abundantly contained in biological material, may form MDA in the TBA reaction mixtures. Hence, its specificity has been strongly questioned for over 30 years [42,43,44,45]. Nevertheless, more sophisticated high-performance liquid chromatographic (HPLC) methods have been developed for the separation of the MDA-TBA adduct from other possible TBA reactants using spectrofluorimetric detection. Consequently, researchers showed that increased serum MDA levels predicted adverse health outcomes in 634 patients with stable coronary artery disease (CAD) [46] and in 774 patients suffering from chronic heart failure (CHF) [47].

F2-isoprostanes belong to stable prostaglandin-like compounds with distinct side-chain structures compared to cyclooxygenase-derived prostaglandins. They are generated in vivo as products of the free radical-induced peroxidation of arachidonic acid [34,35]. Among the group of sixty-four F2-isoprostanes, 8-iso-prostaglandin F2α (8-iso-PGF2α, also known as 8-epi prostaglandin F2α, 8-epi PGF2α) was selected as a quantitative biomarker of oxidative stress [48]. For the first time, the association of oxidative stress with a progression of heart failure in humans (ischemic heart disease (IHD) and/or valvular heart disease) was demonstrated in 1998 by Mallat et al. The scale of the oxidative damage in the heart, expressed by the levels of 8-iso-PGF2α measured in pericardial fluid, was significantly correlated with ventricular dilatation, which was assessed by, among others, the measurement of left ventricular end-diastolic (r = 0.5, *p* = 0.008) as well as end-systolic (r = 0.46, *p* = 0.026) diameters [39]. Moreover, during 17 years of the follow-up study, it was observed that the risk of developing fatal CHD (n = 141) and stroke (n = 109) was 80% higher among postmenopausal women with a urinary concentration of 8-iso-PGF2α included in the highest quartile (the odds ratio was 1.8, and a 95% confidence interval (CI) amounted to 1.1–3.1, *p* = 0.02) [49]. Another case-control study showed that patients with CHD (n = 93) also significantly differed from healthy controls (n = 93) in terms of urinary 8-iso-PGF2α levels (120–193 pmol/mmol creatinine vs. 77–139 pmol/mmol creatinine, respectively, *p* < 0.001). A multivariate analysis indicated that among the five risk factors of CHD (body mass index (BMI), C-reactive protein, high-density lipoprotein (HDL) cholesterol, systolic blood pressure and 8-iso-PGF2α) only C-reactive protein (>3 mg/L, *p* < 0.01) and 8-iso-PGF2α (≥131 pmol/mmol creatinine, *p* < 0.001) had a predictive value of CHD occurrence [33].

In a clinical study, the expression of HNE-modified proteins was measured immunohistochemically in endomyocardial biopsy samples derived from 23 patients with dilated cardiomyopathy. A 5.3-fold greater HNE-positive heart area was demonstrated in these samples compared to those collected from control subjects (n = 13, *p* < 0.0001), which was indicative of the oxidative stress in the myocardium of patients with heart failure [40].

Moreover, other biomarkers of oxidative stress were also found in the course of CVDs, e.g., the products of a nitrosative stress (excessive RNS concentration). They are represented, among others, by 3-nitrotyrosine (3-NT), the nitration product of tyrosine residues in the proteins present in the vessel wall or in the myocardium [3,50,51]. However, the literature provides incoherent data on the prognostic value of 3-NT in the context of CVDs. Using an accurate HPLC with the tandem mass spectrometry method, Shishehbor et al. found significantly higher levels of 3-NT (*p* < 0.001) in the serum proteins of CAD patients (9.1 µmol/mol tyrosine as the median) compared to the group of healthy individuals (5.2 µmol/mol tyrosine) [52]. On the other hand, the ELISA-based measurements indicated a lack of an association between the levels of 3-NT and mortality rates among CAD patients (a follow-up time of 4 years) [53] (Figure 1).

An oxidatively damaged DNA is also observed in people suffering from CVDs. 8-oxo-7,8-dihydro-2′-deoxyguanosine (8-oxodG) was widely assessed as a cellular [54] and urinary [55] marker of DNA oxidative damage. A meta-analytic study performed by Di Minno et al. revealed that 810 patients with CVDs such as CAD, carotid atherosclerosis, PAD and stroke showed markedly higher 8-oxodG levels, both in leukocyte DNA and urine, as compared to 1106 control individuals (standard mean difference (SMD) equal to 1.04, 95% CI: 0.61–1.47, *p* < 0.00001) [56]. A similar meta-analytic review yielded the conclusion that 446 patients with heart failure showed significantly elevated 8-oxodG levels compared to 140 healthy controls (SMD = 0.89, 95% CI = 0.68–1.10, *p* < 0.00001) [57] (Figure 1).

## 4. Exercise Impacts on Redox State

Physical exercise is one of the physiological factors which, through the enhanced generation of ROS and RNS, impacts oxidant–antioxidant balance. The generation of these active derivatives of oxygen and nitrogen is indispensable for the appropriate course of adaptive processes for exercise [58]. However, when ROS are generated in excess, they may lead to the damage of cell components [59,60,61], including lipids, proteins and DNA [62].

The main source of ROS during exercise has not been ultimately defined so far, which results, among others, from the fact that oxidative stress caused by PA may be evoked not only in skeletal muscle but also in other tissues [63]. Initially, what was considered the main source of ROS during exercise was the mitochondrial respiratory chain; however, more recent research demonstrated that skeletal muscle mitochondria produce more ^•^O_2_^−^ in the basal condition than during exercise [64,65]. Under the conditions of PA, what also decreases is the generation of H_2_O_2_ by mitochondria [66]. What is considered as a significant source of ROS during exercise is xanthine oxidase (XO) and NOX. XO is generated under conditions of ischemia from xanthine dehydrogenase. Such a state may be a result of exhaustive physical exercises when blood flow is shunted from many organs and tissues and is redirected to the working muscles. After concluding exercise under the conditions of reperfusion, XO generates ^•^O_2_^−^ and H_2_O_2_ as by-products of xanthine and hypoxanthine oxidation to uric acid [67]. The activation of endothelial XO was proved, for instance, 24 h after a single anaerobic exercise in men and women [68]. A significant source of ROS in skeletal muscles after exercise apart from mitochondria and XO is also NOX. The basic function of this family of enzymes is ROS generation. NOX occurs not only in phagocytic cells but also in other types of cells; among others, in endothelial cells, cardiomyocytes and skeletal muscle [69]. It was demonstrated, among others, that high-intensity interval training in mice is associated with an increase in NOX2 activity in skeletal muscles [70].

PA is also accompanied by an enhanced generation of ^•^NO [71]. The ^•^NO generated in low concentrations in various types of cells has the function of cellular or intracellular signaling molecules engaged, among others, in the regulation of vascular wall tension. Generated in excess, they may indicate destructive properties [72]. Activation was proved of both the iNOS and the eNOS after exhaustive exercise [73]. The source of ^•^NO is also erythrocytes. It is assumed that this is a source of ^•^NO that may play a role in the regulation of local blood flow dynamics during PA [74]. On the other hand, however, the reaction of ^•^NO with ^•^O_2_^−^ leads to the formation of a strong RNS—peroxynitrite (ONOO^−^) [75].

Various types of PA probably elicit different pathways of free radical production [67,76]. The connection between exercise and oxidative stress is extremely complex, depending on the mode, intensity and duration of exercise [77] and the training status of the individual [78]. PA can be differentiated into aerobic and anaerobic (depending on the intensity), acute and chronic (in relation to the frequency) and concentric, eccentric and isometric (in relation to muscle contractions) [67]. It was proved that oxidative stress accompanies both aerobic exercise, especially an intense one [79,80], and anaerobic [80,81]. Aerobic exercise as early as at its beginning leads to the enhanced generation of ROS, whereas anaerobic exercise may induce prolonged ROS generation [82].

Single bouts of exercise lead to oxidative stress [83,84]. As a result of physical training, which is accompanied by repetitive exposure to the enhanced generation of reactive oxygen and nitrogen species, adaptation changes occur, which may protect against oxidative stress. This fact remains in accordance with the so called hormesis theory, according to which low-level exposure to ROS elicits beneficial stress adaptation, whereas too high of a generation of ROS leads to oxidative stress [85]. Tolerance increase to exercise-induced oxidative stress was demonstrated, among others, after aerobic training in overweight/obese adolescent girls [86]. An adaptive increase in antioxidant enzyme activity after training was indicated after 3 months of karate training in elite athletes [87] and in the erythrocytes of handball athletes during 6 months of monitoring [88]. Some of the research also proves that a single bout of exercise confers increased resistance to a subsequent non-exercise oxidative challenge, but only in young people [89].

It was demonstrated that exercise also impacts the existing balance between ROS and RNS in the vascular system. Regular PA improves vascular function, which in part derives from a reduction in cellular ROS and a restoration of ^•^NO bioavailability (a change in the ROS/NO balance to favor ^•^NO) [90].

## 5. Effects of Physical Exercise on Redox Equilibrium in Cardiovascular Diseases

CVDs constitute disorders manifested by a chronic inflammatory state and alterations of the oxidant–antioxidant balance of the oxidative stress character [91,92]. It is common knowledge, however, that muscle activity during physical exercise leads to a significant increase in ROS concentration and improves the antioxidant defense system in the organism [93,94].

Linke et al. examined 23 patients with CHF, aged 56 ± 4, and 12 controls (healthy subjects at an approximate age to the patients). The patients were randomly divided into two groups—the group of people exercising every day for 6 months and the group not exercising during this period of time (sedentary lifestyle). The PA of the patients was diversified; however, it was mainly effort on the cycloergometer for 20 min daily with a heart rate corresponding to 70% of the maximal oxygen uptake. What constituted the test material were the bioptates of the skeletal muscle. SOD, CAT and GPx activity were determined in homogenized muscle samples as well as with nitrotyrosine concentrations. What was observed in patients without division into groups was a significantly lower activity of enzymes and higher concentrations of nitrotyrosine than in the controls with regard to baseline levels. The training, however, caused a significant increase in CAT and GPx activity in the patients, and these values were higher than in the non-exercising subjects, as well as a decrease in the concentration of nitrotyrosine (*p* < 0.05). Thus, it was not proved that PA altered SOD activity in the skeletal muscles of subjects with CHF [95] (Table 1). A lack of an impact of aerobic exercise on SOD was also proved by Fenty-Stewart et al. in the plasma of venous blood in 100 hypertensive patients (50–75 years). However, they observed a positive correlation between SOD activity and 8-iso-PGF2α concentration in urine [96], whereas, Kostić et al. observed a statistically significant decrease in erythrocyte GPx activity after an acute exercise test on the cycloergometer in peripheral blood in 40 hypertensive patients (51.19 ± 8.37 years) [97]. Yu et al. examined dependence between the type and frequency of physical exercise, and redox biomarkers in the plasma of venous blood in persons between 45 and 79 years old with idiopathic arterial hypertension (n = 402), with the use of a control group (n = 1047). The study considered five types of PA, from low intensity to high intensity exercises such as walking/dancing, taiji/yoga, running/biking/climbing, ball sports and gym workouts including swimming. Each type of PA was divided into 3 degrees of performance frequency. The PA of all subjects was therefore rare (0 times/week), moderately frequent (1–3 times/week) or frequent (4–7 times/week). The authors noticed that even the least intensive physical exercise may have had a favorable impact on redox equilibrium (a decrease in MDA and HNE concentration, an increase in SOD activity) in the hypertensive patients, which is also accompanied by improvement with regard to clinical parameters (arterial blood pressure, BMI, glycosylated hemoglobin, triglycerides) if exercises are performed adequately frequently and regularly [98]. On the other hand, in the study on the oxidant–antioxidant balance in peripheral blood in 43 participants with arterial hypertension, aged 60–80, the impact of regular walking performed in a continuous manner (30 min walking at the pace of one step per second, three times a week) and at intervals (3 × 10 min march at the pace of one step per second + 1 min break after each repetition, three times a week) was determined. The group walking in a continuous manner was constituted by 8 males and 14 females, whereas the interval group consisted of 2 males and 19 females. After 12 weeks, the concentration of reduced glutathione (GSH) in both groups was increased, and the concentration of oxidized glutathione (GSSG) was decreased (*p* < 0.05) (both parameters were examined in the whole blood). Additionally, MDA concentrations in plasma in both groups were subject to a statistically significant decrease. The differences between the groups, however, were statistically insignificant with the exception of some clinical parameters (the triglycerides concentration, the ratio of the total cholesterol to HDL and the atherosclerogenic index were lower in the group walking continuously; *p* < 0.05). Thus, the study confirmed the advantageous influence of long-term and regular aerobic exercise of a low intensity on redox equilibrium in the venous blood of hypertensive subjects [99]. Tsarouhas et al., however, determined the total antioxidant capacity (TAC) in the serum of 20 males and 7 females (66.8 ± 13.1 years) with CHF subjected to effort, and 9 males and 3 females (67.0 ± 5.6 years) with the same disease without introducing a training program. All the ill persons were treated with statins, and apart from the TAC, the lipid and glycemic profiles and the TNF-α concentrations in the serum were examined in these patients. The control group constituted 12 healthy males and 5 healthy females (65.7 ± 9.4 years). After 12 weeks of training, an increase in the TAC was observed in the training group in relation to the non-training one. Additionally, the TAC was higher in both groups of persons with cardiac failure in comparison with the control group. The researchers stated that in patients with CHF, daily, moderate, unsupervised physical activity such as walking was able to improve their lipid and glycemic profile with the simultaneous alleviation of the inflammatory state and oxidative stress [100]. In 18 patients with cardiac failure (13 males and 5 females), aged 28–59, venous blood tests were performed after the application of a 30 min and a 45 min effort of low intensity and a 30 min effort of moderate intensity. The MDA concentration in the plasma was determined as well as CAT and SOD activity in the erythrocytes. Moderate physical effort caused the most significant changes; among others, a significant increase in the MDA concentration and CAT activity. Thus, the intensity of the exercises was in this case a more significant factor with respect to evoking physiological effects than the duration of the effort [101]. A long-term and extremely complex study was performed by Raberin et al., who examined the relation between PA and sex, as well as cardiovascular risk factors and redox parameters in peripheral blood in the elderly. Altogether, 1011 healthy subjects were qualified for the study, who were at least 65 years old on recruitment day. The study lasted 10 years in total. During that time, the persons recruited were observed and tested clinically with respect to cardiovascular and cerebrovascular events. In the final year, among 545 persons (318 females aged 75.8 ± 1.2 and 227 males aged 75.8 ± 1.1), redox parameters in the plasma were determined and they were the following: uric acid (UA), ^•^NO (as the sum of nitrite and nitrate (NOx) concentrations), ferric-reducing antioxidant power (FRAP), advanced oxidation protein products (AOPP), MDA, GPx and SOD. The authors concluded in general that biochemical cardiovascular risk factors, including redox parameters, occur in females at a lower level than in males, which may explain the lower percentage of CVDs in older women. It was not observed, however, that PA correlated negatively with the level of cardiovascular risk factors tested both in females and in males [102]. There also exists a study in which redox parameters were determined in saliva. The study encompassed two groups of patients after myocardial infarction, i.e., 21 patients aged 42–71 (67% males, group I) and 21 patients aged 47–79 (81% males, group II). Group I was subjected to physical effort on a cycloergometer, and group II performed breathing and balance exercises. The MDA and TAC concentrations were measured, which were expressed in the form of the 2.2-diphenyl-1-picryl-hydrazyl (DPPH) concentration. The MDA concentration was decreased and the TAC increased in group II (*p* < 0.05). In group I, no statistically significant differences were observed. Thus, it was demonstrated that the form of PA may have a different impact on antioxidant capabilities or on ROS concentration in saliva in patients after myocardial infarction [103]. Lately, comprehensive data on redox balance in individuals with CVDs were provided by Tofas et al., who studied patients suffering from CAD. The patients were submitted to an 8-month training with division into three groups depending on exercise type, i.e., aerobic (n = 15, 61 ± 7 years), anaerobic (weight) (n = 11, 62 ± 8 years) and mixed (aerobic and weight exercises) (n = 15, 64 ± 6 years). The fourth group of patients was the control, not exercising group (n = 15, 64 ± 8 years). The authors revealed that all types of training in CAD patients induced positive changes in the redox equilibrium in their peripheral blood; however, the most remarkable and pronounced alterations were found after aerobic training (compared to pre-training, baseline values). The authors determined the concentrations of thiobarbituric acid reactive substances (TBARS) and protein carbonyls (PCs) in plasma, GSH and GSSG in erythrocytes, as well as the TAC in serum and the CAT activity in erythrocytes. In all patients (all training groups), a statistically significant decrease in the systolic blood pressure was also observed, whereas the diastolic blood pressure decreased in the aerobic and anaerobic groups compared to the control individuals (health and anthropometric measures similar to the non-training patients in the study period; n = 15, 64 ± 8 years). Interestingly, most of the redox status parameters restored in all the CAD subjects close to the pre-exercise values at the end of the 3-month detraining period that was followed immediately by an 8-month training session [104]. The examination of 94 patients (60 males and 34 females, 68.0 ± 14.5 years) hospitalized due to acute heart failure and subjected to an exercise test proved, however, that ROS generation in response to physical effort increases the number of cardiac events. The authors concluded that if antioxidant capacity increases as a result of PA, then we can expect beneficial effects for cardiac failure patients; however, if it does not happen and the ROS concentration significantly increases, the effects are negative [105] (Table 1).

## 6. Pathological Consequences of Oxidative Stress in Cardiovascular System and Relation to Exercise

As mentioned, CVDs are disorders characterized by oxidative stress and inflammation in the myocardium and blood vessels [91,92]. Oxidative stress may be both a primary and a secondary factor of many CVDs [106].

In myocardium, the overproduction of ROS is manifested by the hypertrophy and fibrosis initiated by angiotensin II. This peptide hormone leads to NOX activation. As a result, formed ^•^O_2_^−^ induces redox signaling pathways, the final effect of which is increased protein synthesis, including procollagen I and III [107,108]. Heart disorders induced by oxidative stress may also result from impacts of ROS directly on calcium channels in the sarcolemma (ryanodine receptors) and sarcoplasmic reticulum [109].

Oxidative stress principally triggers specific arterial dysfunction manifested by arterial stiffness—atherosclerosis [106]. ROS, RNS and oxidatively injured molecules induce the expression of inflammatory cytokines and adhesion molecules in endothelial cells and smooth myocytes, leading to the atherogenic remodeling of the vascular wall. Significantly higher concentrations of ROS were reported in acute CAD compared to chronic coronary syndromes (i.e., stable angina pectoris), which also suggests the impact of ROS on atheroma plaque stability [110]. In this case, it is also caused by NOX, which was proven by several clinical studies in human subjects [111]. NOX is activated and induces oxidative stress in the endothelium by angiotensin II-dependent signaling, but also through platelet-derived growth factor (PDGF) and tumor necrosis factor alpha (TNF-α). Hemodynamic parameters influence the NOX activity, as well. Laminar flow decreases NOX activity while a turbulent flow favors the enzyme activity; therefore, arterial hypertension itself may induce atherosclerosis, whereas oxidative stress can be the source of hypertension. ^•^O_2_^−^ decreases ^•^NO concentration to form ONOO^−^, and together with H_2_O_2_ can increase endothelin-1 synthesis and induce endothelial cell apoptosis, leading to vasoconstriction [106]. It was also reported that ONOO^−^ is a significant factor of pathological mechanisms in brain stroke, as it leads to the contraction of the smooth myocytes of the cerebral arterioles [112]. Moreover, increased ONOO^−^ concentrations were found in hyperhomocysteinemia, another important risk factor for endothelial dysfunction, peripheral arterial disease and IHD [113]. Other source of ROS, oxidative stress, inflammation and consequently atherosclerosis may also be XO and LOX. In addition to ^•^NO, oxidatively injured endothelial cells also release significantly lower amounts of other vasodilators (prostacyclin, adrenomedullin, hyperpolarizing endothelial factor) and become too permeable to the oxidized form of low-density lipoproteins (ox-LDL). Oxidative stress disrupts vascular wall function as well as directly acting vascular smooth muscle cells, in which it damages contractile proteins [106].

It should also be mentioned in this section that all the aforementioned molecular mechanisms that lead to atherosclerosis accompany diabetes [106]. Diabetes mellitus (DM) is therefore a pathological condition of the organism that particularly predisposes them to CVDs. Hence, individuals suffering from diabetes are accompanied by hypertension, dyslipidemia, hyperglycemia and insulin resistance [114]. Hyperglycemia, the primary symptom of DM, is also combined with ROS production. The autooxidation of glucose and the non-enzymatic glycation of proteins generates ^•^O_2_^−^. For instance, the glycation may occur directly between glucose and LDL or apolipoprotein B to form advanced glycation end products (AGEs), which induce lipid peroxidation. ROS facilitate that process. Thus, AGEs are pathological changes in the structure of plasma proteins; however, they have their own specific receptors (receptors of AGE, RAGE). The binding of AGEs to RAGE in various cell types (i.e., endothelial cells, smooth muscle cells, macrophages, monocytes and lymphocytes) contributes to ROS production and inflammation dependent on the activation of NOX and nuclear factor kappa B (NF-kB) [106].

The impact of PA on patients with CVDs seems to be unambiguously positive if applied at therapeutic doses. This was confirmed by at least several studies. One of them is the mentioned study of Tsarouhas et al., who reported that in patients with CHF, daily, moderate and unsupervised physical exercise for 12 weeks, such as walking, improved their lipid and glycemic profile with a simultaneous alleviation of the inflammatory state and oxidative stress [100]. In another one, systolic and diastolic blood pressure as well as flow-mediated vasodilation (endothelial function assessment) meaningfully improved after 6 weeks of strength training (knee extensor exercise, three times a week) in six mildly hypertensive men (Stage 1) (71 ± 2 years) [115]. Another study showed that, according to the authors, physical effort improved systolic blood pressure, pulse pressure, central aortic systolic blood pressure and central aortic pulse pressure, and it may delay arterial aging in hypertensive patients (n = 63; age of 40–70 years). The training program included 50 min of exercise, divided into 30 min aerobic exercises and 20 min resistance exercises per session, four times a week for 4 months [116].

## 7. Antioxidant Supplementation—Unconventional Treatment of Cardiovascular Diseases

The primary care level in cardiovascular disease reduction and treatment should cover the following drugs: aspirin, beta-blockers, angiotensin-converting enzyme inhibitors and statins [1]. An especially wide range of action and effectiveness have statins [117]. The anti-inflammatory response of high-dose statins and their beneficial effects on vascular functionality were even proven, which were also accompanied by a decrease in ROS concentration [106]. In this context, antioxidant supplementation also seems to be important since the cardioprotective effects of dietary antioxidants are well known. Indeed, a diet rich in fruits and vegetables results in an increase in serum antioxidant capacity and a decrease in oxidative stress. In contrast, studies on antioxidant supplementation, even those that are numerically significant, have revealed no clear benefit in the prevention and therapy of CVDs. Both short- and long-term clinical trials have failed to consistently support the cardioprotective effects of supplemental antioxidant intake. For these reasons, and also to limit the immoderate consumption of antioxidant supplements, both the US Food and Drug Administration and the European Food Safety Authority have excluded from the package labels any information that could imply the potential health benefits of products with antioxidants [9]. For instance, Ardalani et al. demonstrated that the consumption of *Cynara scolymus* (artichoke) powder as a rich source of phenolic and antioxidant compounds could potentially improve BMI and systolic blood pressure in hypertensive patients [118]. In turn, in the mentioned six men with mild hypertension (Stage 1) (71 ± 2 years), the benefits resulting from exercise (an improved systolic and diastolic blood pressure and flow-mediated vasodilation after a 6-week knee extensor exercise, 3 times/week) deteriorated because of antioxidant supplementation (vitamins C and E, and alpha-lipoic acid). The levels of the parameters returned to the baseline state in the (hypertensive) subjects as a result of this supplementation. Moreover, a single intake of the supplement in a high dose (the documented antioxidant effect in the peripheral blood plasma) before starting the exercise program did not change the levels of the parameters studied [115].

The confusing effects of antioxidant supplementation can be explained in several ways. First, the term “antioxidant” applies to a very wide variety of chemical entities that share only the capability of chemical reduction (the donation of electrons). Dietary antioxidants are substances both with lipophilic and hydrophilic properties; moreover, the size and complexity of antioxidants vary widely. They can be small and simple (e.g., salicylates, ascorbic acid) or large and very complex, like polyphenols (e.g., tannic acid). This heavily impacts their bioavailability, which also depends on metabolism and the distribution of antioxidants in the organism. Bioavailability, which is an active form of the antioxidant in a target place, is a highly complex issue that depends on resistance to digestion and metabolic conversion by the gut microbiome, absorption, metabolism and clearance. Finally, antioxidants are, by definition, rapidly oxidized. Oxidation before or during ingestion might not only abolish antioxidant properties but could also actively promote oxidative stress, depending on the nature of the product formed. Hence, the donation of the electron(s) by the antioxidant is linked to its oxidation, which may lead to the formation of its reactive entity, including radical ones [8]. Moreover, it should be remembered that the effects of antioxidants in the healthy organism, in physiological conditions, may differ compared to pathological conditions, in the course of the disease, especially those characterized by redox disturbances.

## 8. Conclusions

It is widely known that PA positively influences quality of life, improving physical and mental fitness and, consequently, may also reduce the emergence and development of CVDs [119]. CVDs are commonly characterized by glucose metabolic syndrome, oxidative stress and inflammation, whilst PA is known to improve redox status, insulin sensitivity and endothelial function [100]. The effects of PA on oxidant–antioxidant equilibrium can be ambiguous and depend on the duration and intensity of exercise; however, in general, they are positive [95,96,97,98,99,100,101,102,103,104,105] (Table 1). Physical exercise of different intensities (usually of moderate or of low intensity, and long term) is already widely recommended as an aid in the prevention and treatment of CVDs. Exercise is recommended, for example, in the prevention of CAD, and as part of the treatment process in myocardial infarction, coronary bypass graft, heart transplantation and CHDs [93]. With regard to antioxidant supplementation as a potentially new approach to the treatment, it is not recommended due to inconsistent data coming from clinical trials. Probably, the best way to keep the oxidant–antioxidant equilibrium is a supply of antioxidants in a balanced diet, as the literature recommends in the case of exercising people.

## Figures and Tables

**Figure 1 jcm-11-04833-f001:**
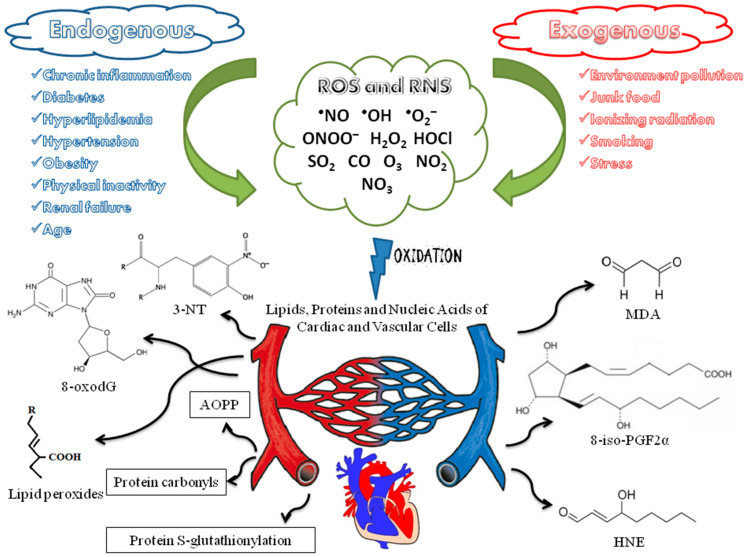
The sources and effects of oxidative stress that promote cardiovascular diseases. ^•^NO—nitric oxide; ^•^OH—hydroxyl radical; ^•^O_2_^−^—superoxide anion radical; ONOO^−^—peroxynitrite; H_2_O_2_—hydrogen peroxide; HOCl—hypochlorous acid; SO_2_—sulfur dioxide; CO—carbon monoxide; O_3_—ozone; NO_2/3_—nitrogen di/trioxide; 3-NT—3-nitrotyrosine; 8-oxodG—8-oxo-7,8-dihydro-2′-deoxyguanosine; AOPP—advanced oxidation protein products; MDA—malondialdehyde; 8-iso-PGF2α—8-iso-prostaglandin F2α; HNE—4-hydroxy-2-nonenal.

**Table 1 jcm-11-04833-t001:** Exercise impacts on redox balance in cardiovascular diseases.

Cardiovascular Disorder	Study Group	Exercise	Material	Outcome	Ref.
Chronic heart failure	Patients exercising and not exercising (n = 23) Controls (n = 12)	aerobic (bpm of 70% VO_2max_), 20 min/day, 6 months	Skeletal muscle	↑CAT ↑GPx ↓NT	[95]
	Patients exercising (n = 27) Patients not exercising (n = 12) Controls (n = 17)	aerobic (moderate), every day, 12 weeks	Peripheral blood	↑serum TAC	[100]
	Patients (n = 18)	a single 30 min bout and 45 min bouts of low intensity, and a single 30 min bout of moderate intensity	Peripheral blood	↑plasma MDA ↑erythrocytic CAT (moderate exercise)	[101]
Acute heart failure	Patients (n = 94)	aerobic and anaerobic, 30 min/day (cardiac rehabilitation), 5 days/week during hospitalization	Peripheral blood	↑serum dROM = ↑no. of cardiac events (poor prognosis)	[105]
Hypertension	Patients (n = 100)	aerobic (50→70% VO_2max_), 20→40 min/day, 3 days/week, 6 months	Peripheral blood, urine	plasma SOD = urine 8-iso-PGF2α (positive correlation)	[96]
	Patients (n = 40) Controls (n = 20)	maximal (a single acute bout)	Peripheral blood	↓erythrocytic GPx	[97]
	Patients (n = 402) Controls (n = 1047)	aerobic (low–high intensity and frequency)	Plasma of peripheral blood	↓MDA ↓HNE ↑SOD	[98]
	Patients (n = 43)	aerobic (low intensity, continuous and intermittent), 30 min/day, 3 days/week, 12 months	Peripheral blood	whole blood ↑GSH ↓GSSG ↓MDA	[99]
Cardiovascular and cerebrovascular events	Patients (n = 545)	Approx. 5 h/day of low-intensity physical activity (<3 METs)	Plasma of peripheral blood	↓redox parameters in females vs. males	[102]
Myocardial infarction	Patients (n = 42)	single aerobic, 30 min (low intensity) (cardiac rehabilitation)—cycloergometer or breathing and balance	Saliva	↓MDA ↑TAC after breathing and balance exercises	[103]
Coronary artery disease	Patients (n = 56)	aerobic, anaerobic or mixed, 50–60 min/day, 3 days/week, 8 months	Peripheral blood	↑GSH ↓GSSG ↓TBARS (aerobic exercises)↓PCs ↑TAC ↑CAT (all types of exercises)	[104]

Abbreviations and symbols: bpm—beats per minute (heart rate); VO_2max_—maximal oxygen uptake; ↑—increased level; ↓—decreased level; CAT—catalase; GPx—glutathione peroxidase; NT—nitrotyrosine; TAC—total antioxidant capacity; MDA—malondialdehyde; d-ROM—diacron reactive oxygen metabolites (mainly hydroperoxides); SOD—superoxide dismutase; iso-PGF2α—8-iso-prostaglandin F2α; METs—metabolic equivalents; HNE—4-hydroxy-2-nonenal; GSH—reduced glutathione; GSSG—oxidized glutathione; TBARS—thiobarbituric acid reactive substances; PCs—protein carbonyls.

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
