# Peer review of "The Impact of Exercise on Redox Equilibrium in Cardiovascular Diseases"

_jcm, 2022, doi:10.3390/jcm11164833_

Round 1

Reviewer 1 Report

The paper of Pawel Sutkowy et al is a review on Redox Equilibrium

Most interesting information are present in the most important paragraphs: The Oxidative stress in cardiovascular diseases, The exercise impact on Redox State and the effects of Physical Exercise  on Redox equilibrium in Cardiovascular diseases

The Introduction,  The Oxidant-Antioxidant Balance in Human Organisms and the Conclusions  are probably excessive for the general equilibrium of the paper

Reviewer 2 Report

-       The proposed article is very interesting and with great potential. I found very useful information that I will gladly cite in my future articles.

-       However, some revisions are needed before it can be accepted for publication:

-       In a world where the sedentary lifestyle is a rule, not an exception, the impact of physical exercise on cardiovascular diseases (no. 1 cause of morbidity and mortality throughout the world) and obesity (increasing prevalence) is a very important topic and you should extend this idea. E.g. – “Risk factors for adiposity in the urban population and influence on the prevalence of overweight and obesity. Experimental and therapeutic medicine. 2020;20(1):129-33.

-       I think a specific section with insights about the impact of physical exercise on endothelial function, arterial stiffness and inflammation in this setting would add to a more exhaustive view of the CVD and the ways to prevent or delay it and its consequences; it would also better fit the scope of the journal, acknowledging that arterial stiffness is believed to be linked with oxidative stress in several disorders by uncoupling of NOS and by oxidative damage to the proteins, lipids, and DNA of vascular endothelial cells and also easily quantifiable. In this paragraph, consider adding doi: 10.1016/j.biopha.2022.113238. Also, you can address briefly the impact of the traditional and unconventional risk factors on the redox system and cardiovascular risk. Consider using DOI: 10.3390/diagnostics10050314.

-       Also, you address the effects of Cynara scolymus on systolic blood pressure, diastolic blood pressure, and flow-mediated vasodilatation (endothelial function assessment), without linking inflammation and arterial function in the previous paragraphs. Consider the following citations to prepare this further chapter: In Vivo, 2016 Jul-Aug;30(4):521-8.

-       Also, in your abstract, you mention that “Antioxidant supplementation is also an important issue since antioxidant supplements still have a great potential regarding their use as drugs in these diseases”, but there is no paragraph about any supplements or their effects, no clinical study. The theme is approached in the “conclusions” section.

-       You should mention the classical and guideline recommendations for ways to decrease inflammation – e.g., statins, before addressing unconventional supplements. In this direction, you could use the following as a reference: doi.org/10.3390/diagnostics10070483 and doi.org/10.1002/fft2.15

-       The English language must be edited throughout the manuscript. You have approximately 400 editing errors and it is difficult to follow the script due to this issue.

Round 2

Reviewer 1 Report

The paper is improved with new text and I suggest to publish this version

Reviewer 2 Report

Authors replied to all the query raised and the paper can now be accepted for publication